# Promoting Spiritual Coping of Family Caregivers of an Adult Relative with Severe Mental Illness: Development and Test of a Nursing Intervention

**DOI:** 10.3390/healthcare12131247

**Published:** 2024-06-22

**Authors:** Tiago Casaleiro, Helga Martins, Sílvia Caldeira

**Affiliations:** 1Faculty of Health Sciences and Nursing, Centre for Interdisciplinary Research in Health, Universidade Católica Portuguesa,1649-023 Lisbon, Portugal; hemartins@ucp.pt (H.M.); scaldeira@ucp.pt (S.C.); 2Escola Superior de Enfermagem São Francisco das Misericórdias, Grupo Autónoma, 1169-023 Lisbon, Portugal; 3Escola Superior de Saúde, Instituto Politécnico de Beja, 7800-111 Beja, Portugal; 4Postdoctoral Program Integral Human Development, Católica Doctoral School, 1649-023 Lisbon, Portugal

**Keywords:** caregivers, coping holistic nursing, nurses, nursing care, psychiatric nursing, spirituality, spiritual coping

## Abstract

Severe mental illness disrupts daily functioning, burdening family caregivers, who often adopt spiritual coping strategies. With comprehensive skills, mental health nurses can promote well-being and mental health. The aim is to develop and test the nursing intervention “promoting spiritual coping” in the family caregivers of home-dwelling people with mental illness. This study was conducted in two distinct stages. Initially, the intervention was developed according to the first phase of the Framework for Developing and Evaluating Complex Interventions. Secondly, the intervention protocol was tested in a mixed-method pilot study. An intervention protocol was developed and tested on ten family caregivers. The intervention comprised three sessions, and before-and-after assessments were conducted. Significant improvements were observed in the outcomes, with caregivers expressing that discussing spirituality and religiosity benefited them. This intervention prioritized the therapeutic relationship of the nurses and family caregivers. The intervention “promoting spiritual coping” was created and evaluated as a suitable approach for mental health nurses to use in a psychotherapeutic context with family caregivers of individuals with mental illness.

## 1. Introduction

Policymakers and healthcare providers have addressed mental health challenges in recent decades but are not always effective and successful [1]. The World Health Organization has revealed that an estimated 13% of the world’s population has a mental health disorder [1]. The impact and severity of mental illnesses vary, and some conditions are categorized as severe or serious mental illnesses [2]. In 2019, approximately 4% of the population had a severe mental illness, including schizophrenia, conduct disorder, bipolar disorder, and major depression [3]. These numbers may be higher globally due to underreporting for various reasons [1].

The recovery-oriented approach to mental health is based on an integrated and comprehensive approach that considers the holistic perspective of the individuals [4]. In addition, the recovery-oriented approach in mental health care also advocates for community-based care, with both formal and informal care [1]. Mental health nurses are specialized healthcare professionals who promote the mental health of individuals and groups and provide care, support, and treatment for individuals with mental health issues.

Caregiving is a challenge for family caregivers who must adapt to the needs of their relatives [5]. These caregivers face a higher probability of developing emotional stress and depressive symptoms [6]. Different coping strategies are used by caregivers, such as cognitive strategies, problem- and/or emotion-focused coping, and religious/spiritual coping [7]. Gall and colleagues proposed a Spiritual Framework of Coping [8]. This seminal work suggests that when facing a stressor, a person appraises the situation influenced by personal factors, meaning in life, spiritual connections, and spiritual coping behaviors [8].

A systematic review explored the spiritual aspects of family caregivers of relatives with severe mental illness, identifying both spiritual needs and spiritual coping strategies used by caregivers [9]. This review revealed that caregivers facing stressful situations turn to the sacred, conduct spiritual/religious practices/rituals, and adhere to a formal religion [9]. Recent research shows that positive spiritual/religious coping strategies positively impact mental health and quality of life [10].

Mental health nurses regularly contact family caregivers, supporting their role and promoting coping behaviors. However, a qualitative study revealed that few mental health nurses provide spiritual care [10]. These professionals acknowledge the lack of training and time required to adequately address spirituality [11].

Moreover, the debate about spirituality in psychiatric and mental health nursing is growing, and a recent study revealed that the attention to spirituality in mental health practice by mental health nurses was influenced by their religiosity, spiritual perspectives, and years of experience [12]. These factors should be taken into account in research conducted in this field.

Since evidence shows the impact of positive spiritual coping, mental health nurses are urged to address spiritual needs and promote coping. This path has challenges and opportunities [13]. There may persist some resistance regarding spiritual care. Nevertheless, there is still a moral and ethical mandate to promote personal resources that impact the well-being, burden, and satisfaction of family caregivers of adults with severe mental illness.

The existence of an intervention to promote spiritual coping would help nurses support their practice and assess the impact of the intervention. Therefore, this study aims to develop and test the nursing intervention ‘promoting spiritual coping’ for family caregivers of a community-dwelling adult relative with severe mental illness.

## 2. Materials and Methods

This study was conducted in two stages. In the first stage, the main goal was to develop an intervention, and a two-step exploratory, descriptive, mixed-methods study was conducted, corresponding to the first phase of the Framework for Developing and Evaluating Complex Interventions [14]. The intervention protocol was tested in a mixed-method pilot study embedded in the framework’s feasibility phase in the second stage. The tests were conducted in Lisbon, Portugal.

### 2.1. Part 1: Development of the Intervention

The guidelines for developing interventions, according to O’Cathain et al. [15], were considered to operationalize the development stage. A spiritual intervention in the Oncologist Assisted Spiritual Intervention Study (OASIS), implemented by oncologists, was found [16]. This intervention consisted of a spiritual inquiry into how people cope with cancer. The current study modified the intervention in OASIS to suit a different context and population.

We engaged with various stakeholders to evaluate how this intervention could be tailored to the specific context. The initial version of the intervention was developed and presented to two separate online focus groups: one comprised family caregivers and the other consisted of professionals, including mental health nurses and researchers in the field. Due to the geographical spread of the participants and COVID-19-related restrictions, online focus groups were deemed appropriate for this study. The first group was composed of a non-probabilistic sample of four family caregivers of adults with severe mental illnesses, and a caregivers’ association facilitated recruitment. The second group included six nurses with extensive experience in practice, teaching, and research who were invited by e-mail. The groups met separately, completing one session each. The sessions took place online in July 2021 and took around 90 min. The groups were asked for their opinion about essential aspects regarding the suitability and pertinence of the intervention, participants, place, procedure, and content of sessions (Figure 1). Both groups received previously, by e-mail, the first version of the intervention protocol, and the interview followed its structure for assessment of the participants.

Deductive content analysis was performed according to Kyngäs and Kaakinen [17]. According to this method, a structured analysis matrix can be defined beforehand. The sessions followed the intervention protocol and allowed the participants to give further comments. The matrix included participants, place, procedure, and session content. The session audio was recorded and transcribed by the lead researcher, then reviewed and validated by an observer, as two researchers were involved in conducting the interviews. The units of analysis were highlighted. The units were then coded using NVivo 13 software [18]. The units were inserted in the predefined categories of the matrix. When needed, subcategories were defined. After analysis, changes were performed in the intervention protocol.

The second version was presented to a panel of nurses, aiming for consensus. Following the TIDieR-PHP checklist [19], a modified e-Delphi was conducted, considering the nurses’ focus group as the first round and one round of questionnaires sent to the participants’ e-mails [20]. They were asked if they agreed with the elements of the intervention: title, pertinence, participants, local, content of session, number and time of sessions, feasibility, and modifiability of the intervention. The answer was on a 4-item Likert scale ranging from totally disagree to totally agree. Participants were offered an open space to comment on each question. The criteria for consensus were adapted from Fink et al. [21]: only those topics that received a mean rating of two or higher were accepted. Additionally, only the issues supported by at least 75% of the participants were adopted, and if more than 65% totally agreed with the item (rate four), it was considered a higher consensus.

### 2.2. Part 2: The Test of the Intervention

A pilot study was conducted to assess the intervention’s results in a small group and test the processes [22]. The hypothesis that the intervention ‘promoting spiritual coping’ does not affect spiritual coping, quality of life related to physical and mental health, or the burden of caregivers of relatives with severe mental illness living at home was tested. The intervention consists of three sessions, one-to-one, delivered across six weeks. The intervention in the current study took into account both the Spiritual Framework of Coping by Gall and colleagues [8] and the Nursing Theory of Self-Transcendence by Reed [23].

#### 2.2.1. Participants

A non-probability convenience sampling method was used to select ten participants. The number of participants is minimal when the aim is to test the process and assess its acceptance [24]. The inclusion criteria were defined as adult family caregivers of an adult with severe mental illness living at home. A family member is considered to have a first- or second-degree kinship. All individuals who wished to participate in this study and demonstrated their capacity to consent were included. Family caregivers of adults with the following mental illnesses and comorbidities were excluded: substance use disorder, dementia, and neurodevelopmental disorders, along with family caregivers who obtained a score of 0 on the Brief RCOPE-PT instrument when completing the questionnaire.

The mental health community team provides care to around 75 persons with severe mental illness. The participant’s enrollment process started with a mental health nurse from the mental health community team making the first contact with the family caregivers. Consent was obtained to provide their contact information to the principal investigator. After this step, the principal investigator (first author) made an initial phone call to each participant and initiated the intervention following the protocol.

#### 2.2.2. Instruments

In this study, four instruments were used for data collection. Firstly, a sociodemographic questionnaire was developed for this phase of the study. Then, the Brief RCOPE-PT scale (13 items), already validated in European Portuguese for family caregivers of people with health problems [25]—Brief RCOPE-PT—was used. This is a two-dimension instrument. The positive spiritual coping subscale’s score ranges from 7 to 28, and the negative spiritual coping ranges from 6 to 24. The highest value represents the frequent use of the coping strategy. The second version of the 12-item Short Form Health Survey—SF-12v2 [26]—was used to assess the health-related quality of life. This instrument assesses not only physical but also mental health. The scores range from 0 to 100. A score of 0 indicates the lowest level, and 100 is the highest. The brief version of the Informal Caregiver Burden Assessment Questionnaire (QASCI) [27] was used to assess the burden. In this instrument, higher scores represent a higher burden, and scores range from 0 to 100.

#### 2.2.3. Data Collection

The researcher who conducted the intervention applied the four instruments before the intervention and immediately after the intervention.

After the intervention, an open-ended question was aimed at obtaining participants’ experiences during the intervention. The use of the mixed methodology in the feasibility test is based on the philosophical perspective of critical realism, which allows for a positivist approach, evident in the quantitative study, with the constructivist approach that collects participants’ experiences throughout the intervention [28].

#### 2.2.4. Statistical Analysis

IBM SPSS Statistics 28 software was used to analyze the quantitative data. A descriptive analysis was performed, and the internal validity of each scale, both pre- and post-intervention, was calculated using Cronbach’s alpha homogeneity test. Additionally, the median and 25–75 percentiles for each instrument’s pre- and post-intervention results were calculated. Individual changes in the scores of each instrument were also calculated.

A non-parametric Wilcoxon test for two paired samples was used to test the hypotheses, and the significance level was set at *p* ≤ 0.05. A non-parametric test was chosen because the conditions for parametric tests were not met as the sample size was less than 30 participants [29].

#### 2.2.5. Content Analysis

Recordings were transcribed, and inductive content analysis was performed following the guidelines of Kyngäs [28] to analyze qualitative data obtained through an open-ended question.

First, units of analysis were defined as words or phrases. Categories were then created by combining two or more units of analysis [30]. The text was coded using NVivo 13 software (QSR International Pty Ltd., Doncaster, Australia).

### 2.3. Ethical Considerations

All procedures were performed according to ethical standards, and the institutional ethics committee approved this study. All participants provided written informed consent for participation in this study, the recording of the audio of the focus groups, and the open-ended questions in the pilot test. In addition, measures were taken to preserve participants’ anonymity and confidentiality through the attribution of codes to each participant and the removal of aspects that could identify the participants.

The researchers and nurses involved in the research have their own religious and spiritual beliefs. Efforts were made to ensure that the research team’s spiritual or religious backgrounds or beliefs did not put respondents under pressure to engage in the research in particular ways. This was achieved through regular meetings and supervision after contact with the participants.

## 3. Results

### 3.1. Part 1: Development of the Intervention

The focus groups included ten participants: four family caregivers (P1–P4) and six nurses and researchers (P5–P10).

After coding the transcripts with NVivo, the analysis matrix was completed [17] with information from both focus groups. The pre-established categories were participants in the intervention, place, procedure, and content. The item “Procedure” was divided into two subcategories: the number of sessions and follow-up sessions. The item “Content” was split into two subcategories: introduction and the concepts of spirituality vs. religiosity. In Table 1, it is possible to identify some units of analysis that were categorized, including the procedure category, with more comments.

Through a deductive content analysis, it was possible to confirm the aspects regarding the participants in the intervention and the place of intervention in the first version of the protocol. Nevertheless, comments and suggestions regarding the number of sessions and their content made the need for changes in the intervention evident. The focus group of family caregivers considered the intervention pertinent and useful, and their comments were in line with the intervention’s characteristics. The focus group of experts raised more questions regarding the procedure and the content. The research team considered it important to obtain consensus on the new version of the intervention.

A new version of the intervention has been developed (Figure 2). The second version consisted of three sessions. Some questions regarding experience as a caregiver were added. The terms “spirituality” and “religion/religiosity” were also clarified.

The six experts were then invited to respond to a modified e-Delphi. All the participants answered positively. The questionnaire included ten closed-ended questions with a four-item Likert scale.

The criteria for consensus were obtained during the first round. All the topics were accepted; more than 75% of the participants rated each item 3 or 4. Five out of ten received higher accordance, as more than 65% of the experts rated them as 4. In addition to the closed question, it provided an open space for the experts’ comments.

The consensus was generally high concerning the relevance of the intervention, but a noteworthy comment questioned its applicability to individuals who do not use spiritual coping strategies. A stipulation was incorporated into the protocol to address this concern, specifying that the intervention ends if an individual responds defensively or declines to engage with this spiritual aspect. Additionally, one expert emphasized the importance of caregivers’ prior interactions with psychiatric and mental health nurses, as a pre-existing therapeutic rapport is deemed essential. Consequently, it was included in the protocol that caregivers should already have some level of contact with the mental health team. Regarding the intervention’s dosage, there was a consensus recommending a minimum session length of 45 min, and one expert suggested incorporating a follow-up session. Concerns also arose regarding session closure, prompting the addition of a directive in each session and at the intervention’s conclusion, emphasizing the nurse’s role in reviewing the session and procedure to enhance the caregiver’s autonomy.

At the end, the intervention consisted of three sessions, conducted by one mental health nurse, the first two spaced two weeks apart, with the final session scheduled one month later. Each session lasted between 45 and 60 min. An interview guide was designed to assist the nurse, but modifications to the topics could occur depending on the caregiver’s needs (Appendix B). The interview included questions about the experience of caring for a relative with mental illness and the use of spirituality as a way to cope with stress.

This modified e-Delphi assessed the validity of the intervention ‘promoting spiritual coping of family caregivers of a relative with severe mental illness’.

### 3.2. Part 2: The Test of the Intervention

The pilot study was conducted between June and October 2022 with a sample of 10 family caregivers of individuals with severe mental illness living at home. The mental health nurse from the community mental health center referred the family caregivers and made the first contact. The three intervention sessions occurred at the caregivers’ chosen locations: at home and in other public spaces. The protocol was followed throughout the intervention with all participants.

Table 2 provides an overview of the sociodemographic characteristics of the participants. The average age of the individuals being cared for in this sample was 48.6 years, with a standard deviation of ±10.56. The majority (6 out of 10) of them had a mood disorder, while psychotic disorders (2 out of 10) and personality disorders (2 out of 10) were also identified.

During the intervention, it was possible to identify those caregivers engaged in private and communitarian religious or spiritual practices, such as individual prayer, contact with nature, and participation in religious services and sacraments.

The instruments used in the pilot study revealed a good internal consistency in both the pre- and post-intervention: Brief RCOPE (T0 α = 0.71; T1 α = 0.76); Brief RCOPE positive religious coping subscale (T0 α = 0.94; T1 α = 0.94); Brief RCOPE negative religious coping subscale (T0 α = 0.77; T1 α = 0.72); QASCI (T0 α = 0.66; T1 α = 0.70); SF-12v2 Physical Health Summary (T0 α = 0.93; T1 α = 0.95); and SF12v2 Mental Health Summary (T0 α = 0.89; T1 α = 0.90). The individual changes in the scores (see Appendix A) reveal that 8 out of 10 participants had positive changes, and 2 out of 10 had no change in the Brief RCOPE positive religious coping subscale; 6 out of 10 had positive changes, and 4 out of 10 with no change in the Brief RCOPE negative religious coping subscale. Regarding the burden, 8 out of 10 participants changed positively, whereas two remained with the same score. In the Physical Health Summary of SF-12v2, seven participants had a negative change, and three improved. Inversely, eight participants had a positive change in the mental health summary and two had a negative change.

Table 3 contains the summary of the 25th percentile (Q1), median, and 75th percentile (Q3). In the subscale of Positive Spiritual Coping of Brief RCOPE and in the Mental Health Summary of SF-12v2, the median increased, showing an enhancement in these outcomes. The median of the QASCI reveals a decrease in burden. The results regarding the Physical Health Summary of SF-12v2 show slight changes in all quartiles. The Negative Spiritual Coping subscale results show that the median did not vary but there was a change in both Q1 and Q3. In all the outcomes, there were changes in both Q1 and Q3, suggesting a widespread impact, affecting both the lower and higher ends of the score distribution. The interquartile range showed mixed results, with a small increase in the variability of the scores in the Positive Spiritual Coping subscale and the Mental Health Summary, and a small decrease in the other outcomes. Additionally, a summary of Wilcoxon’s signed rank test results is provided as a supplement. The results also demonstrate that the intervention enhances the quality of life related to mental health in these participants (*p* = 0.047, Z = −1.988 ^a^). However, the hypothesis that the intervention improves the quality of life related to physical health was rejected (*p* = 0.285, Z = −1.070 ^b^).

Regarding the outcome of positive spiritual coping, the findings show that the intervention increases positive spiritual coping (*p* = 0.011, Z = −2.539 ^a^). Similarly, the results show that negative spiritual coping decreases after the intervention (*p* = 0.024, Z = −2.264 ^b^).

Furthermore, the family caregiver burden decreased with the intervention (*p* = 0.011, Z = −2.536 ^b^).

In the third session, an open question was posed regarding experience during the intervention. After a careful review of the documents, 21 units were identified. Considering these similarities, five categories have emerged (Table 4).

## 4. Discussion

In this study, a specialized nursing intervention was developed that engaged stakeholders and focused on the essential aspect: the therapeutic relationship between the nurse and the person, in this case, the family caregiver. The research was centered on nursing practice, contributing an additional intervention that can address the needs of an often-neglected population.

In the development phase of the current study, the caregivers considered the intervention relevant and important as part of the attention given to the caregivers. Additionally, they reinforced that this intervention needs to target family caregivers with prior contact with a mental health team. Given all the above, this population is overloaded at different levels and needs a complex response from several areas [31]. Thus, addressing spiritual issues cannot be disconnected from other needs and challenges associated with the caregiving experience. Embracing this holistic perspective is a step further in delivering care in the mental health field [4].

The development of this intervention considered the different stakeholders involved in the care delivery. This iterative process changed a two-session intervention strongly focused on the spiritual dimension into a three-session intervention with broader questions.

The sample characteristics closely resemble the findings from a European study [32]: predominantly female caregivers over 45, with a notable proportion being elderly (5 out of 10). One aspect that was most evident in this sample was the aging of the caregivers, which we highlight is a public health concern due to the associated vulnerabilities [33]. Regarding the caregivers’ spirituality and religiosity, the majority are Roman Catholics, which is similar to what was found in a survey on religion in the geographical area of Lisbon, Portugal [34].

Conducting a pilot study with a small sample allows for the identification of aspects of the intervention to be changed. Data collection is assumed to be preliminary and allows for estimating the effectiveness of the intervention in a future randomized study. Hypotheses were tested with pre- and post-intervention evaluations. This initial data collection is considered one of the elements of the guidelines for the development of interventions [15].

Regarding the quality of life related to health, it was established that the intervention in this sample does not increase the quality of life related to physical health. However, effectiveness was observed regarding the quality of life related to mental health. In the intervention protocol, some issues are linked to the caregiving experience and the use of other coping strategies in addition to those related to spirituality. Relating the content analysis results to caregivers’ experience in this intervention, this was demonstrated in the “interconnection of spirituality with caregiving experience” category. In this category, caregivers reveal that the intervention is linked with spirituality and other aspects of their lives.

The findings of the positive spiritual coping, measured through the positive religious coping subscale of Brief RCOPE-PT, show that the intervention increased the spiritual coping values. In addition, the intervention decreased the negative spiritual coping assessed through the instrument. The caregivers revealed that the intervention allowed for an increase in knowledge and spiritual coping strategies, as evidenced through content analysis. Furthermore, it was possible to verify a decrease in caregiver overload, considering the total value of the scale.

The positive spiritual coping subscale results point to the use of spirituality to a greater or lesser degree. If spirituality is a vital resource, the intervention can help maintain this resource. In a qualitative study on elderly spirituality, it was identified that the best way to provide spiritual support depends on how the older person expresses that same spirituality, which is the starting point [35].

Now, focusing our attention on the content analysis of the responses to the open-ended question at the end of the intervention, we can still find some reservations in talking about spirituality issues because family caregivers do not feel an openness to talk about the subject or consider that this is a field of nurse intervention. This finding aligns with what was identified in Cone and Giske’s [36] study, where mental health professionals themselves identify difficulty in addressing the subject.

In general, the caregivers showed satisfaction with the intervention and considered it relevant, although with the caveat that exploring this topic should not be isolated from other aspects of caregiving. This aspect had already emerged previously in the caregiver focus group. These findings support the value of addressing the physical, emotional, social, and spiritual aspects of a person’s life in the context of their mental health recovery-orientated plan [4].

When analyzing the procedure, a psychoeducational dimension became evident, which involves teaching coping strategies to deal with emotions resulting from role performance. However, this psychoeducational dimension is based on an established relationship with psychotherapeutic foundations. In a recent systematic review and meta-analysis of studies conducted with family caregivers, the effect of different mental health interventions on their mental health was widely identified [37]. Therefore, the efficacy was recognized in studies with psychoeducation, psychosocial interventions, multicomponent cognitive behavioral therapy, mindfulness, and interventions with support groups [37].

We align with a perspective advocating that mental health and psychiatric specialist nurses are competent enough to conduct psychotherapeutic interventions in mental health [38]. The latest statement is reinforced by authors who proposed a model of psychotherapeutic intervention in nursing with an integrative approach of brief or intermediate duration, based on the therapeutic relationship in different settings and with a session duration between 45 and 60 min [39].

## 5. Conclusions

### 5.1. Limitations

In the first part of this study, the stakeholders were invited to provide feedback on the pertinence and suitability of the intervention. The intervention protocol received insightful comments from the focus groups, but the number of participants was restricted. The participation of a larger and more diverse group would enrich the discussion and note some aspects that may have been neglected. For instance, spiritual support workers or chaplains could have been part of the focus group.

The pilot test was the first step towards a future randomized clinical trial to assess the intervention’s effectiveness with control and experimental groups. Apart from the small sample, it is also identified as a limitation of the assessment immediately after the intervention. Nevertheless, the impact of the intervention over time is unknown.

It is possible to identify not only a predominance of Christian denominations in this sample but also an aging group of participants. This represents a limitation as the impact of the intervention on caregivers with other characteristics, such as being from other cultures, backgrounds, and ages, is unknown. Nurses must acknowledge spiritual and religious diversity with increased migration across countries [40].

Concerning the intervention, a significant limitation is the difficulty in isolating the independent variable. When delivering the intervention, the mental health nurse develops a therapeutic relationship with the person so that other factors may confound the results. Another challenge is the difficulty in standardizing the intervention, as it depends on the therapeutic relationship established. Still, one of the strategies to reduce this limitation could be a training activity for professionals who implement this intervention in clinical practice.

### 5.2. Implication for Practice

The intervention ‘promoting spiritual coping’ for family caregivers of people with severe mental illness can be considered a brief psychotherapeutic intervention in clinical practice carried out by a mental health nurse.

Throughout this intervention in clinical practice, nurses can facilitate caregivers of people with severe mental illness to reframe their experiences and find meaning. Through self-transcendence, caregivers acknowledge vulnerability and boundaries, with nurses leading the self-transcendence process. This fosters connections with self, others, nature, and the transcendent, aligned with caregivers’ beliefs. Given the risk of isolation, nurses must attentively identify and provide therapeutic support to help the caregivers. The assistance for family caregivers is often neglected and overlooked, as the health systems focus on the person with mental illness. If they were less burdened and had a better quality of life, family caregivers would be more able to maintain the role of caregiver.

## Figures and Tables

**Figure 1 healthcare-12-01247-f001:**
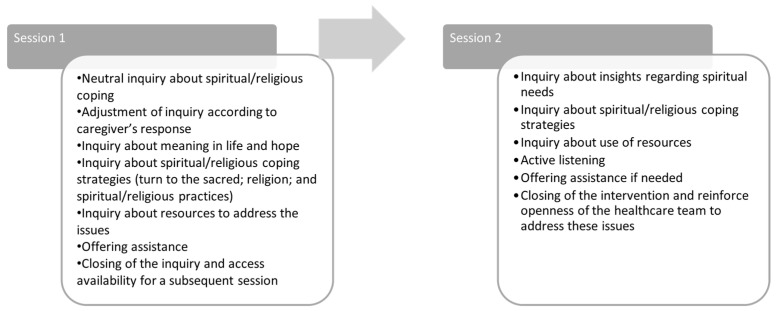
Session plans of the first version of the intervention.

**Figure 2 healthcare-12-01247-f002:**
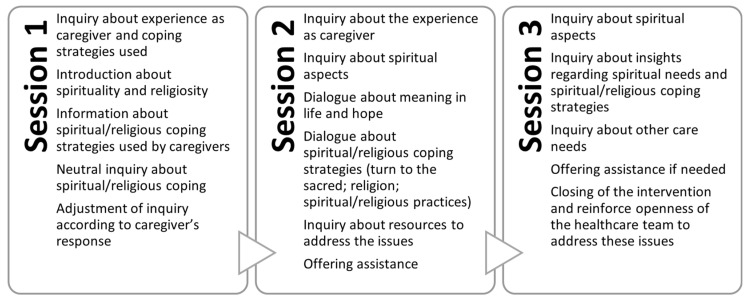
Session plans of the second version of the intervention.

**Table 1 healthcare-12-01247-t001:** Categorization of findings in focus groups.

Categories		Number of Codes	Examples
Participants in the intervention		8	“[it was important that] this theme was part of the standard intervention attributed to the healthcare team, and it could be attributed to nurses” (P2) “This intervention could be part of nursing care as a complement of the relationship established with the persons” (P3)
Place		5	“in what concerns with the place, I think it depends on the participants, the place where they feel comfortable” (P2)
Procedure	Number of sessions	16	“if the aim is to capacitate for spiritual coping, then it is needed more [sessions]” (P7)
	Follow up	7	“I think that two sessions may be enough, but there is a risk that the caregivers feel a connection with the nurse and look for further support” (P2) “It feels that we look at the persons, we raise a bunch of questions, talk with him or her once again, and then we vanish” (P7)
Content	Introduction	10	“[in the beginning] to open more, to ask a broader question (…) how the person is dealing with the experience … so … don’t direct immediately to spiritual coping” (P7)
	Spirituality vs. religiosity	6	“not focus that much in the interventions directed to religiosity but also give caregivers some suggestions regarding spirituality” (P6)

**Table 2 healthcare-12-01247-t002:** Characteristics of the pilot test’s participants.

		N
Gender	Female	9
	Male	1
Caregiver’s age (µ = 61.8)	Under 45	1
	Between 45 and 54 years	1
	Between 55 and 64 years	3
	Over 65 years	5
Marital status	Single	2
	Married/Civil partnership	6
	Widower	1
	Divorced	1
Education	1st cycle of primary education (4th year)	3
	Secondary education (12th grade)	6
	Higher education (bachelor’s, bachelor’s, master’s, or doctorate)	1
Degree of kinship	Father/mother	5
	Son/Daughter	1
	Sibling	3
	Partner	1
Level of care	Permanent	6
	Regular, but not permanent	4
Cohabitation	Yes	6
	No	4
Religiosity/spirituality	Spiritual and religious	8
	Spiritual but not religious	2
Religion	No religion	2
	Christianity (Roman Catholic church)	6
	Jehovah’s witnesses	2

**Table 3 healthcare-12-01247-t003:** Summary of median and percentiles pre- and post-intervention.

	Pre-Intervention Q1, Mdn, Q3	Post-Intervention Q1, Mdn, Q3	Statistics *p*
Brief RCOPE-PT—Positive Spiritual Coping	13.00, 16.5, 23.75	14.75, 18.5, 26.25	0.011 *
Brief RCOPE-PT—Negative Spiritual Coping	7.00, 8.00, 10.25	6.75, 8.00, 9.25	0.024 *
SF-12v2—Physical Health Summary	55.84, 58.5, 59.88	55.68, 57.92, 59.18	0.285
SF-12v2—Mental Health Summary	52.04, 53.55, 54.62	53.27, 54.39, 56.54	0.047 *
QASCI	38.84, 49.11, 54.46	36.61, 45.54, 50.45	0.011 *

Legend: Q1—25th percentile; Q3—75th percentile; Mdn—median; * Indicates a statistically significant change.

**Table 4 healthcare-12-01247-t004:** Inductive content analysis.

Examples of Analysis Units	Number of Units of Analysis	Category
“[These issues] are not always talked about by nurses and other health professionals” (P2) “Normally, these subjects are not spoken” (P4) “Initially, I thought it was strange for a nurse to talk about these things” (P9)	3	Taboo
“For me, it was positive to talk about this topic but also to be able to vent about my concerns” (P5) “What I am now talking to you also helps me in other aspects, in my day-to-day life” (P6) “But I keep thinking I miss support in other aspects of my life.” (P10)	4	Interconnection spirituality with caring experience
“At first I didn’t realize what spirituality was, but now I know it’s more than going to church because I’m away” (P2) “I end up knowing other things, and spirituality is what also moves me to have more strength in the day-to-day, to help me in the day-to-day, which is not easy...” (P3) “I realized how good this can be to take care of me... Being in nature, helping others, having moments for myself. I need to take care of myself” (P9)	3	Increase of knowledge and strategies
“To talk about it, it is always good; it always does us much good spiritually in our life” (P1) “It is quite useful” (P3) “It is important for nurses to talk about this topic from the moment it helps us” (P6)	3	Relevance of the intervention
“It is very important to me, I enjoyed talking, talking to you” (P1) “I enjoyed talking about these topics” (P2) “It was nice to know they cared about that... I felt more understood.” (P7) “Talking about it does me good” (P8)	8	Satisfaction with the intervention

## Data Availability

Dataset available on request from the authors.

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
