# Peer review of "Promoting Spiritual Coping of Family Caregivers of an Adult Relative with Severe Mental Illness: Development and Test of a Nursing Intervention"

_healthcare, 2024, doi:10.3390/healthcare12131247_

Round 1

Reviewer 1 Report

Comments and Suggestions for Authors

I'm grateful for the opportunity to review this paper. Thank you for your research on this important topic.

This paper reports on the development and pilot of an intervention tool for mental health nurses working with family caregivers of adults with mental illness. Its findings affirm the value of opening conversation about and supporting the spiritual resources of those in difficult caregiving roles. The authors fairly acknowledge the limitations of their study. The intervention tool provided in the appendix will be a useful resource for those working in the field.

I noticed only a few spots requiring attention:

Line 178 – the words ‘cared for’ are repeated

Line 282 – incomplete sentence?

Lines 357-359 – This sentence is confusing to me. Please clarify.

Author Response

Reviewer 1

I'm grateful for the opportunity to review this paper. Thank you for your research on this important topic.

This paper reports on the development and pilot of an intervention tool for mental health nurses working with family caregivers of adults with mental illness. Its findings affirm the value of opening conversation about and supporting the spiritual resources of those in difficult caregiving roles. The authors fairly acknowledge the limitations of their study. The intervention tool provided in the appendix will be a useful resource for those working in the field.

I noticed only a few spots requiring attention:

We want to thank the reviewer for this comment and the suggestions provided.

Line 178 – the words ‘cared for’ are repeated

Due to other revisions, this paragraph was moved and the words were corrected (Lines 94-95)

Line 282 – incomplete sentence?

The sentence was rephrased to make it more precise and complete (lines 294-295).

Lines 357-359 – This sentence is confusing to me. Please clarify.

It was rephrased. We acknowledge that focus groups included a small number of participants. A larger group would eventually get more insights into the intervention protocol. (Lines 369-374)

Reviewer 2 Report

Comments and Suggestions for Authors

An interesting article on a very relevant topic! It's reasonably well-written, but there are some areas that could use clarification, particularly to make it clearer.

Abstract

The abstract comes across a bit messy. It initially focuses on family caregivers and then suddenly shifts to nurses in the third sentence, without immediately establishing the connection for the reader. Perhaps this link could be clarified from the outset. The introductory part may be shortened - to be more concise - and provide a bit more explanation about the content of the intervention and/or the outcome measures. Was it a qualitative or quantitative study? What method was employed? Etc. The discussion section could then be shortened again.

Introduction

The introduction starts off very well and is clearly written. However, from R55 onwards, the coherence decreases, possibly due to a lack of use of words like 'because', 'since', 'however', etc. For example, why does it matter in the context of the current study that nurses' attention to spirituality depends on their background and experience? And is the lack of training a motivation for this program - is it a form of training? And why suddenly bring up the moral and ethical aspect? Perhaps it could be better linked to the utility of spiritual coping, which is described earlier? In short, there could be a clearer progression towards the purpose of the study.

Materials and Methods

R80/81  The authors could add to the sentence that ‘the intervention’ here refers to the intervention developed in the current study. R83-R90 can have some grammatical improvements for clarity.

R83-R93 Ah, understood. Here's the translation: I'm still not quite following. Initially, I thought there was one group consisting of family members and another of professionals, or did both groups meet twice? And was it actually an open interview, or was the intervention already provided beforehand for assessment? Was the interview about spiritual coping or about the usability of the intervention, or both? Or was it a mixed group with both family members and healthcare professionals?

R105 How were they validated?

R111 Could the researchers provide some examples of the questions here? They write about topics and issues but as a reader I have no idea where it is all about.

R125-126 I wonder, was a power calculation not needed? The sample is really small for a quantitative study

R127 possibly more consistency in using past tense

R131-132 An interesting decision to exclude participants with a score of 0... What is the motivation behind this? Did the researchers expect that this group would not benefit from spiritual coping?

R133-137 I really miss the information about the content of the intervention and how it is delivered. Is it a one-on-one session? Or group-based? Is it delivered multiple times or just once? Is there a specific psychological theory behind it? Perhaps this information is in a supplement, but as a reader, I would also like some information in the methods section, even if it's a brief summary.

R143-144 What exactly were these time points?

R168 What kind of measures?

Results

R177 Aha, here the matter about the groups is clarified – possibly this sentence and a bit more clarification could be moved to the method section?

R180 The researchers sometimes talk about professionals in general, but I think I have read these are all nurses is not it? Could they not better talk about nurses?

R193 Formulation is a bit odd

R202 Why only the experts? When reading I wonder: do the researchers aim to have conclusions about the feasibility and usefulness of the intervention by answers of the experts or the family caregivers. Could the researchers clarify this somewhere?

R238 ‘possible to identify’ is a bit strange in formulation

R252 I did not see that hypothesis, the researchers stated in R 121 that the intervention would not affect spiritual coping, quality of life, or burden – they either do not mention physical health and state the hypothesis the other way around

Discussion

Overall, the discussion is still a bit scattered; that is to say, there are many valuable components, but I still miss the coherence. Sometimes it is unclear whether it concerns the quantitative or qualitative results and whether it involves the nurses or the caregivers, etc. I advocate for rewriting the discussion with more coherence. In the discussion it is not necessary to repeat the method stuff. Below are a few additional points.

R276 the formulation ‘it was possible to identify’ is used regularly, but I guess the sentence does not add much here.

R284 Indeed que population is quite old, but was it a random sample? They were also quite religious, was this by purpose? I am also interested how the age of the caregivers could have influenced the results.

R328: nurses or family caregivers?

Conclusion

R359 I indeed wondered what the time between the two measurements was

R365-366 How many different nurses delivered the interventions? Is that mentioned somewhere?

A few additional questions: Why is it relevant to focus on family caregivers? Are they also at risk for mental disorders or something else? Does it help to relieve the mental health care system? And who actually finances the assistance for them? How does this assistance for family caregivers relate to the importance of spiritual support for the patients themselves? Are they aware of this? Do they also receive assistance?

Comments on the Quality of English Language

The english is generally fine, but some sentences can be improved. 

Author Response

Reviewer 2

An interesting article on a very relevant topic! It's reasonably well-written, but there are some areas that could use clarification, particularly to make it clearer.

The authors thank the reviewer for the encouragement and insightful comments. We addressed each topic individually and considered that the process helped us strengthen our arguments.

Abstract

The abstract comes across a bit messy. It initially focuses on family caregivers and then suddenly shifts to nurses in the third sentence, without immediately establishing the connection for the reader. Perhaps this link could be clarified from the outset. The introductory part may be shortened - to be more concise - and provide a bit more explanation about the content of the intervention and/or the outcome measures. Was it a qualitative or quantitative study? What method was employed? Etc. The discussion section could then be shortened again.

Thank you for the comment.

We edited the abstract according to your suggestions. (R 15-27)

Introduction

The introduction starts off very well and is clearly written. However, from R55 onwards, the coherence decreases, possibly due to a lack of use of words like 'because', 'since', 'however', etc. For example, why does it matter in the context of the current study that nurses' attention to spirituality depends on their background and experience? And is the lack of training a motivation for this program - is it a form of training? And why suddenly bring up the moral and ethical aspect? Perhaps it could be better linked to the utility of spiritual coping, which is described earlier? In short, there could be a clearer progression towards the purpose of the study.

The reviewer's questions helped us reflect deeply on our argument. We changed the section to make the argument more solid. (eg. R 54-59; R 62-64; R 65-68)

Materials and Methods

R80/81  The authors could add to the sentence that ‘the intervention’ here refers to the intervention developed in the current study. R83-R90 can have some grammatical improvements for clarity.

The whole paragraph was rephrased for clarity( R 87-98).

R83-R93 Ah, understood. Here's the translation: I'm still not quite following. Initially, I thought there was one group consisting of family members and another of professionals, or did both groups meet twice? And was it actually an open interview, or was the intervention already provided beforehand for assessment? Was the interview about spiritual coping or the usability of the intervention, or both? Or was it a mixed group with both family members and healthcare professionals?

We rewrote it to make it clearer: two groups met separately, one with family members and the other with healthcare professionals. They received the first version of the intervention protocol beforehand. (R 87- 104)

R105 How were they validated?

Information was moved to this section to make it clear (R 111-113).

R111 Could the researchers provide some examples of the questions here? They write about topics and issues, but as a reader, I have no idea what they are all about.

A sentence was added to provide this information (R 121-123).

R125-126 I wonder, was a power calculation not needed? The sample is really small for a quantitative study

The sample size was recognized as a limitation. However, the aim was to test the intervention and its feasibility. For this reason, the generalization of these results isn’t adequate.

R127 possibly more consistency in using past tense

An effort was made to make it more consistent.

R131-132 An interesting decision to exclude participants with a score of 0... What is the motivation behind this? Did the researchers expect that this group would not benefit from spiritual coping?

We had many thoughts when deciding on this aspect. When scoring 0 on this scale, it means that the question does not apply to that person. For this reason, and keeping in mind that researchers were cautious not to force the theme onto the participant, we decided not to include an eventual participant with this score.

R133-137 I really miss the information about the content of the intervention and how it is delivered. Is it a one-on-one session? Or group-based? Is it delivered multiple times or just once? Is there a specific psychological theory behind it? Perhaps this information is in a supplement, but as a reader, I would also like some information in the methods section, even if it's a brief summary.

A summary was provided in the introduction to subchapter 2.2. (R 134-137)

R143-144 What exactly were these time points?

The information was added (R 158)

R168 What kind of measures?

The information was added (R 182-183).

Results

R177 Aha, here the matter about the groups is clarified – possibly this sentence and a bit more clarification could be moved to the method section?

Following your suggestion, clarification was provided in the methods section (R 91-93).

R180 The researchers sometimes talk about professionals in general, but I think I have read these are all nurses is not it? Could they not better talk about nurses?

Whenever “professionals” meant exclusively nurses, the word was replaced (e.g. R 118 and 120).

R193 Formulation is a bit odd

The sentence was rewritten for clarity (R 204-206).

R202 Why only the experts? When reading I wonder: do the researchers aim to have conclusions about the feasibility and usefulness of the intervention by answers of the experts or the family caregivers. Could the researchers clarify this somewhere?

The justification was added to the document (R 207-211).

R238 ‘possible to identify’ is a bit strange in formulation

The paragraph was changed.

R252 I did not see that hypothesis, the researchers stated in R 121 that the intervention would not affect spiritual coping, quality of life, or burden – they either do not mention physical health and state the hypothesis the other way around

We didn’t mention that quality of life was related to physical and mental health. The appropriate changes were made in Subchapter 2.2 (R 254-257)

Discussion

Overall, the discussion is still a bit scattered; that is to say, there are many valuable components, but I still miss the coherence. Sometimes it is unclear whether it concerns the quantitative or qualitative results and whether it involves the nurses or the caregivers, etc. I advocate for rewriting the discussion with more coherence. In the discussion it is not necessary to repeat the method stuff. Below are a few additional points.

Thanks for the suggestion. We appreciated, and we deleted some of the method issues in the discussion (R. 292-295).

R276 the formulation ‘it was possible to identify’ is used regularly, but I guess the sentence does not add much here.

The sentence was changed (R 294).

R284 Indeed que population is quite old, but was it a random sample? They were also quite religious, was this by purpose? I am also interested how the age of the caregivers could have influenced the results.

The sample wasn’t randomly selected. However, its characteristics were in line with the findings from other studies with caregivers. They were also pretty religious. These aspects were outlined in the limitations section (R 379-382).

R328: nurses or family caregivers?

It was clarified (R 341).

Conclusion

R359 I indeed wondered what the time between the two measurements was

This aspect was added to the methods section (R 158).

R365-366 How many different nurses delivered the interventions? Is that mentioned somewhere?

It was mentioned that the same nurse (the lead researcher) delivered the intervention (subchapter 2.2 R 149-151)

A few additional questions: Why is it relevant to focus on family caregivers? Are they also at risk for mental disorders or something else? Does it help to relieve the mental health care system? And who actually finances the assistance for them? How does this assistance for family caregivers relate to the importance of spiritual support for the patients themselves? Are they aware of this? Do they also receive assistance?

Thank you for all the questions and comments. They were immensely helpful in enhancing our work. Some of these questions were answered at the beginning and end of the article (E.g. R 401-404).

Reviewer 3 Report

Comments and Suggestions for Authors

Dear authors, I have read your manuscript with interest. I believe it deserves publishing but after revisions. Please focus more on the intervention and the qualitative part then to the test results. my comments are below:

Major comments:

Introduction:

Please explain for less experienced readers what is a mental health nurse, as this is not a profession in all countries.

Methods:

The sentence „The hypothesis that the intervention promoting spiritual coping121

does not affect spiritual coping, quality of life, or the burden of caregivers of relatives with 122

severe mental illness living at home was tested. please explain why did you have a null hypothesis?

I don't understand what was the intervention, please explain in more details. The intervention is the core of your work and has to be reproducible.

Ethical considerations – please state what ethical committee approved your study and give the number of approval.

Results:

The intervention has to be explained in more details in the text (I know it is all in supplement). You state 3 sessions – what is their duration, structure, was it conducted by one person, etc??? How was the sample chosen and why is it so small?

The sample size is really small (n=10) and test are not useful. What would be useful is to present individual change in scores and report how many people had positive/negative differences or no differences after the intervention and explain it in relation to the clinical norms of those tests.

Table 3 is completely uninformative at has to be replaced with the table where differences are presented; you can leave the test but medians and 25-75 percentiles are welcome, not ranks. Also, create a supplement and present individual data and differences.

Discussion:

The discussion has to be focused on the development of the intervention not to the test results where you describe 10 caregivers. There are numerous quantitative studies investigating caregivers and this information is well known. Therefore condense this part of the text and focus more on the intervention and qualitative results which are interesting.

Minor comments:

Introduction:

p.1. l 34 – please indicate whose data (WHO)

p.2 , l 50 – it is not clear what are the resuts of this systematic review you quote, as if a parto f the sentence at the end of the sentence is missing

Methods:

Statistical analysis – you can add a subtitle

Results:

3.2.  – please don't use percentages instead of 60% write 6/10 or 6 out of 10.

Discussion:

Please do not report in percentages as you have only ten participants.

Author Response

Dear authors, I have read your manuscript with interest. I believe it deserves publishing but after revisions. Please focus more on the intervention and the qualitative part then to the test results. my comments are below:

The authors thank the reviewer for the encouragement and insightful comments. We addressed each topic individually.

Major comments:

Introduction:

Please explain for less experienced readers what is a mental health nurse, as this is not a profession in all countries.

The information was added (L.43-45)

Methods:

The sentence „The hypothesis that the intervention ‘promoting spiritual coping’ 121

does not affect spiritual coping, quality of life, or the burden of caregivers of relatives with 122

severe mental illness living at home was tested.“ – please explain why did you have a null hypothesis?

Although we acknowledge that ten participants in the pilot test don’t allow us to generalize the results, we wanted to test all the procedures. As so, we had a null hypothesis as in a proper effectiveness study. 

I don't understand what was the intervention, please explain in more details. The intervention is the core of your work and has to be reproducible.

We added information regarding the intervention (e.g in subchapters 2.2 (L 134-137) and 3.1 (L 242-243). The appendix provides the content of the sessions.

Ethical considerations – please state what ethical committee approved your study and give the number of approval.

This information is on lines 421-423.

Results:

The intervention has to be explained in more details in the text (I know it is all in supplement). You state 3 sessions – what is their duration, structure, was it conducted by one person, etc??? How was the sample chosen and why is it so small?

An effort was made to detail the information about the intervention. (l 136-139 and l.241- 248)

As stated in lines 140-143, the sample was purposely small as the aim was to test the intervention and procedures. We acknowledge the limitations of this choice. 

The sample size is really small (n=10) and test are not useful. What would be useful is to present individual change in scores and report how many people had positive/negative differences or no differences after the intervention and explain it in relation to the clinical norms of those tests.

Table 3 is completely uninformative at has to be replaced with the table where differences are presented; you can leave the test but medians and 25-75 percentiles are welcome, not ranks. Also, create a supplement and present individual data and differences.

These aspects were discussed in the research team. The size of the sample was considered a limitation. We value your suggestion about the individual scores and maybe a multi-case study would be an option. We chose this statistical test to assess if the intervention was effective.

Regarding the data in Table 3. As the sample of this study is less than 30 participants, non-parametric statistical tests is recommended. Therefore, the data is presented in ranks, sum of ranks and mean ranks.

Discussion:

The discussion has to be focused on the development of the intervention not to the test results where you describe 10 caregivers. There are numerous quantitative studies investigating caregivers and this information is well known. Therefore condense this part of the text and focus more on the intervention and qualitative results which are interesting.

We made changes in the discussion section, adding elements (e.g  l. 307-310) about the development of the intervention and removing some information about the caregivers (eg sentences from the paragraph 311-317 or 338-339).  

Minor comments:

Introduction:

p.1. l 34 – please indicate whose data (WHO)

The information was added

p.2 , l 50 – it is not clear what are the resuts of this systematic review you quote, as if a parto f the sentence at the end of the sentence is missing

The sentence was rephrased (l 55-57)

Methods:

Statistical analysis – you can add a subtitle

We thank your suggestion. After discussion, the authors considered that adding this subtitle would imply create other subtitles (such as data collection, participants, etc). For this reason, we suggest keep it as it was.  

Results:

3.2.  – please don't use percentages instead of 60% write 6/10 or 6 out of 10.

We changed as suggested (l 260-262)

Discussion:

Please do not report in percentages as you have only ten participants.

We changed it as suggested.

Reviewer 4 Report

Comments and Suggestions for Authors

Dear Authors,

Thank you for submitting your very interesting study to the journal for consideration. Your work looks at a spiritual coping intervention for caregivers who have an adult relative living with severe mental illness. This is an aspect of care that is often overlooked and needs to be addressed.

Parts of the article are well written, while other sections need reviewed in order to help and guide the reader.

This is a two phased study, phase one - designing an intervention and phase two - testing the intervention on ten caregivers.

The article begins with a clear abstract outlining what will be addressed.

There is a well written introduction and background, helping to set the scene. You present some global figures on mental illness but you do not mention the gross under-reporting, due to limited resources in many countries.  

Phase one involved focus group discussions with care givers and 'experts', I suggest that the caregivers are the real experts, perhaps you could replace the 'experts' with 'professionals' or another term. You give some idea of what the focus group added to the intervention but this could be clearer. Where there similarities/differences between the two groups (caregivers/experts)? Who were these professionals with extensive experience? This is very vague. Where spiritual support workers/chaplains involved? Some more detail would help.

The intervention appears to be focused on three sessions but what these sessions were could be made clearer to the reader. Where did these sessions take place, of what did they consist. Where is the link with spiritual coping? guide the reader.

There does not seem to be any mention of the training required of the nurse to carry out this intervention. Was it carried out by one nurse? 

Four instruments were used pre and post the intervention.

The quotes in table 4 are quite insightful and give a richer sense of the intervention impact. I wonder whether this table could be written as a section of text, with some context to each quote, this might enrich your work.

It appears that the intervention contributed to the quality of life linked to  caregiver's mental health but not to their physical health. However, this is quite vague.

What does the term 'decreased negative spiritual  coping' mean?

Is a spiritual coping intervention the same as a psychotherapeutic intervention? You seem to think it is. I am not sure I would agree.

The limitations are clear.

As you say, this intervention cannot be seen in isolation from the physical, emotional and social component of the caregiver.

I think the conclusion and implications for practice would be stronger if the above points where addressed.

I think this could be a very interesting contribution to the journal. 

Author Response

Reviewer 4

Dear Authors,

Thank you for submitting your very interesting study to the journal for consideration. Your work looks at a spiritual coping intervention for caregivers who have an adult relative living with severe mental illness. This is an aspect of care that is often overlooked and needs to be addressed.

Parts of the article are well written, while other sections need reviewed in order to help and guide the reader.

This is a two-phased study: phase one - designing an intervention and phase two - testing the intervention on ten caregivers.

The article begins with a clear abstract outlining what will be addressed.

Dear reviewer, we would like to thank you for your words of encouragement and your valuable comments.

We edited the paper and improved the writing.

Thank you for the kind comment.

There is a well written introduction and background, helping to set the scene. You present some global figures on mental illness but you do not mention the gross under-reporting, due to limited resources in many countries.  

This noteworthy aspect was added to the introduction (R 38-39).

Phase one involved focus group discussions with care givers and 'experts', I suggest that the caregivers are the real experts, perhaps you could replace the 'experts' with 'professionals' or another term. You give some idea of what the focus group added to the intervention but this could be clearer. Where there similarities/differences between the two groups (caregivers/experts)? Who were these professionals with extensive experience? This is very vague. Where spiritual support workers/chaplains involved? Some more detail would help.

The word expert was replaced. The authors agree that the true experts, by experience, are the caregivers. The term “expert” follows the methodological use of the Delphi method.

The differences between the two groups were explicitly added in the methods section (L 91-95)

Unfortunately, spiritual support workers or chaplains could not be included, adding to the limitations (l. 373-374).

The intervention appears to be focused on three sessions but what these sessions were could be made clearer to the reader. Where did these sessions take place, of what did they consist. Where is the link with spiritual coping? guide the reader.

An effort was made to guide the reader through the changes in subchapters 2.2 (L 134-137) and 3.1 (L 242-243). The appendix provides the content of the sessions.

There does not seem to be any mention of the training required of the nurse to carry out this intervention. Was it carried out by one nurse? 

As stated in subchapter 2.2, the intervention was carried out by the principal investigator, a mental health nurse. This study aimed to test the intervention. Nurses will need to be trained to implement it on a larger scale.

Four instruments were used pre and post the intervention.

The quotes in table 4 are quite insightful and give a richer sense of the intervention impact. I wonder whether this table could be written as a section of text, with some context to each quote, this might enrich your work.

Thank you for the suggestion. However, the data in the table allows for systematically organizing data, allowing for a clear and structured visualization for the reader. In addition, it permits a better understanding of the data, as it presents information categorized logically and easily.

It appears that the intervention contributed to the quality of life linked to  caregiver's mental health but not to their physical health. However, this is quite vague.

This result is drawn from the instrument SF-12 and the statistical test applied. We do not have more data to specify the changes.

What does the term 'decreased negative spiritual  coping' mean?

The sentence was rewritten (L 327).

Is a spiritual coping intervention the same as a psychotherapeutic intervention? You seem to think it is. I am not sure I would agree.

We had that same discussion, and we considered that promoting spiritual coping could be included in a psychotherapeutic intervention. When developing this intervention, we aimed to provide nurses with an intervention protocol they can use with family caregivers.

The limitations are clear.

As you say, this intervention cannot be seen in isolation from the physical, emotional and social component of the caregiver.

I think the conclusion and implications for practice would be stronger if the above points where addressed.

I think this could be a very interesting contribution to the journal. 

We want to thank the reviewer for the comments and suggestions, and we hope the changes performed throughout the document adequately address the concerns raised. Thank you for your time and input.

Round 2

Reviewer 3 Report

Comments and Suggestions for Authors

The manuscript is revised,but there are still some unanswered questions that were marked as major and therefore my decision is still the same. I give here precise isntructions how to revise results so there will be no confusion.

Author Response

The manuscript is revised, but there are still some unanswered questions that were marked as major and therefore my decision is still the same. I give here precise instructions how to revise results so there will be no confusion. Unanswered comments:

Thank you for your detailed feedback and for providing precise instructions on how to revise the article. We have made the requested changes and addressed the major concerns previously marked in the revised manuscript.

1. How was the sample chosen? - please tell us how many caregivers are in that area and explain how did you choose this 10.

This aspect was clarified in the methods section. (L. 151-154)

2. The sample size is really small (n=10) and test are not useful. What would be useful is to present individual change in scores and report how many people had positive/negative differences or no differences after the intervention and explain it in relation to the clinical norms of those tests.

- Please create a supplement where individual results of each 10 people are presented and measure the pre-intervention and post-intervention difference. Count how many had better results in each test. Explain the results in text (number of positive changes, negative changes, no changes). You have to create at least 5 tables (Brief, sf-12 and QUASCI).

As suggested, five tables were created as a supplement, and the results were explored in the text (L 294-301).

3. Table 3 is completely uninformative at has to be replaced with the table where differences are presented; you can leave the test but medians and 25-75 percentiles are welcome, not ranks. - Please create a table as instructed. Ranks are not informative at all, we don't know the median of the group or can compare those results to the ones previously obtained by different authors. I have

Your perspective and your suggestions helped us to have a new look at our work.

We thank you for that and proceeded as suggested:

- table 3 was replaced by the one suggested (L 302 – 313). We just changed the organization of the information (Quartile 1, Median, Quartile 3)

The table with the Wilcoxon test was provided as supplementary information (See supplement).

New comments:

Methods: Please add an explanation about the range of results in each questionnaire: min-max results and if there are any norms also.

In the methods section, subsubsections were created. One is about the instruments and how to interpret the results. (L 159-170)

Results; - Please remove percentages from table 2 -

The percentages were removed.

Table 3. Why the total score for sf12 is not presented? -

The SF12v2 data calculate two summary component scores, Physical Component Summary Score (PCS) and Mental Health Component Summary Score (MCS). For this reason, only the summary scores are presented.

Please add subtitles in the methods, participants also, not only statistical analysis

As suggested, the material and methods section was restructured, and subsubsections were added. (L 182-190)